# A Case Series Study of Help-Seeking among Younger and Older Men in Suicidal Crisis

**DOI:** 10.3390/ijerph18147319

**Published:** 2021-07-08

**Authors:** Pooja Saini, Jennifer Chopra, Claire A. Hanlon, Jane E. Boland

**Affiliations:** 1School of Psychology, Faculty of Health, Liverpool John Moores University, Liverpool L3 3AF, UK; j.chopra@ljmu.ac.uk (J.C.); c.a.hanlon@ljmu.ac.uk (C.A.H.); 2James’ Place Liverpool, Liverpool L8 7NG, UK; jane@jamesplace.org.uk

**Keywords:** suicide, men, help-seeking, engagement, community-based intervention

## Abstract

Due to the continuing high suicide rates among young men, there is a need to understand help-seeking behaviour and engagement with tailored suicide prevention interventions. The aim of this study was to compare help-seeking among younger and older men who attended a therapeutic centre for men in a suicidal crisis. In this case series study, data were collected from 546 men who were referred into a community-based therapeutic service in North West England. Of the 546 men, 337 (52%) received therapy; 161 (48%) were aged between 18 and 30 years (mean age 24 years, SD = 3.4). Analyses included baseline differences, symptom trajectories for the CORE-34 Clinical Outcome Measure (CORE-OM), and engagement with the therapy. For the CORE-OM, there was a clinically significant reduction in mean scores between assessment and discharge (*p* < 0.001) for both younger and older men. At initial assessment, younger men were less affected by entrapment (46% vs. 62%; *p* = 0.02), defeat (33% vs. 52%; *p =* 0.01), not engaging in new goals (38% vs. 47%; *p* = 0.02), and positive attitudes towards suicide (14% vs. 18%; *p* = 0.001) than older men. At discharge assessment, older men were significantly more likely to have an absence of positive future thinking (15% vs. 8%; *p* = 0.03), have less social support (45% vs. 33%; *p* = 0.02), and feelings of entrapment (17% vs. 14%; *p* = 0.02) than younger men. Future research needs to assess the long-term effects of help-seeking using a brief psychological intervention for young men in order to understand whether the effects of the therapy are sustainable over a period of time following discharge from the service.

## 1. Introduction

With over 800,000 people dying by suicide each year worldwide [1], suicide remains a significant yet preventable public health risk. Suicide is a leading cause of mortality for young men in most high- and middle-income countries [2]. Over the past decade, the rate of suicide among young men has statistically increased by 27.9% from 6.1 deaths per 100,000 males to 7.8 deaths per 100,000 [3]. For young males in England, hospital admissions because of self-harm have also significantly increased during the same period by 6.8% (from 196.8 admissions per 100,000 in 2012 to 2013, to 210.2 admissions per 100,000 in 2018 to 2019) [3]. The reasons for a change in the national rate of suicide are complex and will rarely be due to one factor alone. Among young people, for example, adverse childhood experiences, academic pressures, bereavement, self-harm, and exposure to harmful online content will all be important [4].

Suicide risk factors specific to young men include psychiatric illness, substance misuse, ethnic origin, lower socioeconomic status, rural residence, and single marital status [2,5]. Population-level factors include unemployment, social deprivation, and media reporting of suicide [2]. To date, research and policy concerning young male suicide risk have tended to focus on the male tendency to conceal mental distress as the barrier to engaging with interventions. While young men can be educated regarding known risk factors for suicide, these risk factors may occur in varying levels, and suicidal behaviour is not limited to those in identified high-risk groups [6]. From a preventive standpoint, due to high suicide rates among young men, there is a need to understand more of the complexity that places younger men in particular at risk [7]. For example, there is a need to understand more of the psychological characteristics and mechanisms, such as entrapment, helplessness, social isolation, and self-esteem that regulate the dynamics of suicide in young individuals [8,9,10,11,12,13]. Previous studies have highlighted a dynamic model for how young men were entrapped in what they may have experienced [14]. Others have noted that the suicidal act was understood as a “triggered event” related to a previous significant event close in time, such as a breakup with a partner or a separation from family [4,14]. However, these studies were retrospective and relied on third party information from those bereaved by suicide, and to date, there is limited research on men who are in contact with services for suicidal ideation [15].

In many countries, the main focus in suicide prevention strategies is the identification and treatment of mental disorder, depression in particular, for people who may be at risk of dying by suicide [16]. However, there is growing evidence that many suicides are not preceded by symptoms of serious mental disorder [17,18]. Furthermore, a major challenge for suicide prevention is that most people who take their own lives are not in contact with mental health services at the time of their death and often do not seek help from any health professionals at the time they actually make the decision to end their life [17,18]. Due to high suicide rates and low rates of help-seeking in suicidal crises, young men, in particular, are of great concern [15,19,20]. From a preventive perspective, there is an alarming call to go beyond the medical model and explore the signs that might indicate danger of suicide in the near term, including resistance against help-seeking among young men [21,22,23]. Psychological autopsy studies have highlighted the association of mental health disorders for many youth suicides; however, they also report low rates of contact with mental health services prior to death [24,25]. Young men have been reported to seek help from primary or specialist healthcare services less than other population groups prior to suicide [20,26]. The reluctance to seek help when faced with symptoms of emotional or psychological distress has been highlighted in the wider literature [15,27].

In terms of emotional difficulties and help-seeking, men seem to have higher thresholds than women, particularly when focusing on gender roles [28]. Previous research has highlighted that many young men hid their difficulties and emotions from family and friends prior to dying by suicide [29,30]. For example, population-based analyses of health care contacts among Canadian suicide decedents in Toronto reported that while 10% of men (*n* = 200) were not connected to any form of medical care in the year prior to their suicide, over 60% (*n* = 1792) had accessed professional mental health care in the year before their death [31], thus highlighting the potential lost opportunity to engage men in treatment prior to their death [32,33]. Men with suicidal histories described fragmented mental health care pathways that included negative experiences of service providers and health care systems to the extent that most participants’ service use was involuntary [32]. Men reported discomfort disclosing emotional distress to therapists, and sometimes when desperation prompted their self-disclosures about suicidality, they reported judgement, mislabelling, and an underestimation of their needs. This lack of interest and decreased therapeutic alliance tended to influence men to discontinue therapy and/or opt to self-manage their mental illness [32]. However, over time, a generational change and shifting values have been noted, as some men whose culturally informed ‘strength-based’ masculine ideals to disclose mental illness, vulnerability, and the acceptance of help have changed [33,34,35]. Other studies have operationalised such ideas as affirming men’s help-seeking as courageous and strength-based in tailoring male suicide prevention programs accordingly [2]. However, some of these studies are outdated, and shifts have been seen in more recent research.

Previous findings suggest that existing suicide prevention services are incompatible with the needs and preferences of men who are experiencing suicidal distress [26,36,37,38]. While significant challenges have remained for identifying men at risk of suicide, the importance of building effectual services for addressing men’s self-silencing and enhancing awareness of their own risk status is vital for reducing male suicide. Moreover, the limits of current services confirm the need to better diversify and tailor services to bridge men’s health inequities amid norming men’s mental health help-seeking [39,40], particularly within community settings [6]. Community-based suicide prevention initiatives can enhance the potential of providing support to young men in crisis, through specific provisions for developing openness in communication and responsiveness and improved education about suicide risk. Recent research has suggested that men particularly have the need to receive support from a trusted individual, preferably in an informal setting [41]. Facilitating rapid access to a community-based centre could overcome the problems associated with poor help-seeking behaviours and communication of suicidal distress among young men. It would also offer the desired informal setting, which would be a much-needed lifeline to men in suicidal crisis that cannot be provided by conventional primary care or emergency departments, where it has been reported that young men may have felt judged and not listened to prior to suicide. Brief psychological interventions have been shown to be effective in the prevention of suicide [42,43]. While some have reported promising findings, such as the Atlas wellbeing pilot, which reported positive improvements in psychological wellbeing including anxious mood and stress [44], there remains a paucity of evaluative studies. Subsequently, a knowledge gap exists between what researchers and practitioners know reliably works in suicide prevention interventions for men in a community setting.

### The James’ Place Model

James’ Place is a community-based service delivering a clinical intervention for men in crisis based in North West England. During the planning of the service, experts-by-experience, including those who had been bereaved by male suicides and men who had previously been suicidal, were involved in all discussions with stakeholders who had expertise in suicide prevention (e.g., commissioners, health professionals, local authorities, and academics). Additionally, a focus group discussion was conducted with eleven men who had previously been suicidal in order to gain their views on how the service should be and what they would have found beneficial. The main outcomes of these discussions were the importance of natural furnishings; a homely, safe environment; and an outdoor space to go to if they were feeling claustrophobic. The service took note of the men’s feedback and implemented these changes into the design and look of the building. Men were invited back to view the service and reported that their ideas had been applied, and many wished they had been able to use a similar service when they had previously been in crisis [43]. James’ Place delivers an intervention based on three theoretical models: Interpersonal Theory of Suicide [45], the Collaborative Assessment and Management of Suicidality CAMS; [46], and the Integrated Motivational-Volitional Theory of Suicide IMV; [47,48]. All three approaches include working alongside the suicidal person to coproduce effective suicide prevention strategies and safety planning [49]. Partnerships across the city enabled men to be referred to James’ Place from emergency departments, primary care, mental health services, local universities, or via self-referrals. Clients were offered the James’ Place model that included approximately ten sessions of therapy; however, the number fluctuated depending on each client’s individual needs. Experienced therapists who were trained to deliver the James’ Place model provided sessions.

The model includes nine sessions of therapy in three lots of three. The first three sessions are given over the first week and typically involve the assessment formulation stage where therapists assess the risk of the men, in a collaborative way, with a safety plan. The first stage is about managing the risk, making sure the men are safe and engaged in the talking therapy. The ‘Lay your Cards on Table’ model is introduced within the first three sessions to aid conversation and visually display how the men are being affected by their suicidal thoughts. The middle part lasts over 10 days and is more person centred. The therapists may conduct brief psychological interventions if someone is struggling with negative beliefs about themselves or unhelpful cognitions. This may include behavioural activation, relaxation with someone who is really struggling with anxiety, or sleep hygiene. The final three session will typically consist of relapse prevention and going through a very in-depth safety plan, making sure that the men know the progress they have made and that they know what has actually helped them. That could include using the cards: taking all the cards out and looking at what has been useful and what has not been useful, looking at that person’s early warning signs and what is a sign for them when they are going downhill again, and planning with them for that scenario, so a lapse is less likely to turn into a relapse.

More detailed outcomes for the service are available in two published reports [43,50].

This paper aimed to evaluate an innovative suicidal crisis for younger men using the James’ Place service. Uniquely, this service, the first of its kind in the UK, delivers a clinical intervention within a community setting for men in suicidal crisis.

## 2. Materials and Methods

### 2.1. Participants

This is a case series study of young men experiencing a suicidal crisis who had been referred to the James’ Place Service between 1 August 2018 to 31 July 2020 (*n* = 546). Referrals came from emergency departments (ED), primary care, universities, or self-referrals. Ethical approval was granted by the Liverpool John Moores University (Reference: 19/NSP/057), and written consent was gained from men using the service at their initial welcome assessment.

### 2.2. Clinical Records

Clinical records were compiled for each of the men referred into the service. Sociodemographic information and precipitating factors for men help-seeking in suicidal crisis were entered into the clinical records from the completed referral forms that were received from referral services (EDs, General Practitioners [GP], universities, etc.) or by the men for self-referrals. Therapists also completed this information where it was missing and where it was deemed appropriate to collate for the men. The CORE-34 Clinical Outcome Measure (CORE-OM); clinician assessment of psychological, motivational, and volitional factors; number of sessions (engagement with sessions); reasons for drop out; and referrals out were included within the clinical records. Records were stored on an online computer system that was updated by service administrators or therapists before or following a session with a client. The data collected are described in more detail below.

### 2.3. CORE-34 Clinical Outcome Measure (CORE-OM)

The CORE-OM is a client self-report questionnaire, which was administered at the beginning and end of the therapy. The therapist gave the questionnaire to the men at their first session and then after their final session. To maintain confidentiality, men were given the option to complete the questionnaire alone or with their therapist in the room. As part of the therapy, therapists did use the questionnaire as a source of information to aid the initial session. Men have given feedback to the therapists [43,50] that they found this helpful, as it helped them to speak about specific points they had answered and sometimes they found it was an ‘icebreaker’. The CORE-OM was sent to the men using an online link for the end of the treatment questionnaire; thus, the men would usually not be with the therapist at that time. Men were also able to contact the centre after filling in their questionnaire at the end of their treatment if they found it distressing in any way. The questionnaire included 34 items about how they had been feeling over the last week, using a 5-point Likert scale ranging from ‘not at all’ to ‘most of the time’. The 34 items cover four dimensions, subjective well-being, problems/symptoms, life functioning, and risk/harm, producing an overall score called the global distress (GD) score. A comparison of the pre- and post-therapy scores offers a measure of ‘outcome’ (i.e., whether or not the client’s level of distress has changed and by how much).

CORE-OM data are routinely collected by psychological therapy services [51]. Recent research has shown that participants find the CORE-OM useful in assessing psychological distress and progress within treatment [52]. The measure shows good reliability and convergent validity with other measures used in psychiatric or psychological settings [53,54]. Connell et al. [55] published benchmark information and suggested a GD score equivalent to a mean of 10 or above was an appropriate clinical cut-off, demonstrating a clinically significant change, while a change of greater than or equal to 5 was considered reliable.

### 2.4. Assessment of Psychological, Motivational, and Volitional Factors

A range of psychological, motivational, and volitional factors that play a role in suicidality was assessed by the therapist during each session. These were informed by leading evidence-based models of suicidal behaviour, which the James’ Place model is based upon. Therapists received training on how these factors should be assessed and recorded by the service. When men discussed factors such as ‘feeling trapped’, ‘being a burden’, or ‘lack of ‘social support’ these would be recorded in their clinical record at each session. In addition, the referrer to the service and the precipitating factors to the suicidal crisis were recorded. With regard to the precipitating factors, therapists were trained on recognising the outcomes to reduce subjectivity and recorded this information at the time of consultation, thus reducing recall bias. Feedback was sought from men about their experience of the service once discharged via an anonymised survey that was completed and returned to the service via post or placed in a box at the centre. It should be noted that some of the secondary outcomes are subjective due to referrer or therapist interpretation. Additionally, the men often completed the CORE-OM in the presence of the therapist which may have caused further interpretation bias. However, within service evaluation interviews, both men and therapists reported how the environment played a positive role in the therapy, as men felt comfortable and at ease when attending sessions [43,50].

### 2.5. Engagement with Sessions

Engagement with sessions includes men attending a welcome assessment and at least one session of therapy sessions. Those who only attended a welcome assessment and did not attend any further sessions were classed as incomplete. The number of sessions was determined by the number of times the men attended for therapy sessions. This was recorded within their clinical records.

### 2.6. Data Analysis

Our sample size was predetermined based on the number of men using the service in the first two years since opening. Data were analysed using SPSS 26 [56]. To examine client outcomes, repeated measure general linear models were used to compare pre- and post-treatment data. Magnitude of effect sizes (r) were established using the Cohen criteria for r of 0.1 = small effect, 0.3 = medium effect, and 0.5 large effect.

Descriptive statistics were carried out to illustrate the sociodemographics of the sample and the precipitating factors for men attending in suicidal crisis. Manovas were conducted to establish differences between groups on the core outcome measures at the beginning and end of the treatment. Young men were defined as 18–30 years old, and the older men category relates to men 31 years old and over. Age 30 was chosen as the cut-off age, as the previous literature within suicide prevention has described young people as 18–30 [6,14].

For referrals, these were coded as secondary care (mental health practitioners, crisis and urgent care, ED), primary care (GPs, nurses, support workers, improving access to psychological therapies [IAPT], occupational health, and student wellbeing services), self-referrals (individual/family member), and other (voluntary organisations and charities).

Clinical records from the service were available for the entire sample. Researchers had access to the data, extracted the information, and stored it in excel spreadsheets and SPSS software files to complete the analysis. However, the records only captured entries made in clinical records; unrecorded clinical activity or missing information from referral documents was therefore unavailable. For the purposes of this study, only the presence of each factor within each client’s clinical records was used for the analysis. It is possible this strategy may have led to underestimation of some factors: for example, sexual orientation. Where clients are noted to have completed the intervention, this indicates that the therapy sessions were attended but does not indicate that the discharge CORE-OM questionnaire was filled in.

## 3. Results

Between 1 August 2018 and 31 July 2020, James’ Place received 546 referrals from ED, primary care, universities, or self-referrals. Of those, 417 (76%) attended a welcome assessment, and 337 (81%) went on to engage in therapy (see Figure 1). For those who did not attend the welcome assessment, the reason was usually no response when the men were followed-up, or some said they were not feeling suicidal anymore. The mean age was 34 years (range 18–66 years). Of the 337 men, 161 (48%) were aged between 18 and 30 years (mean age 24 years, SD = 3.4).

The speed with which men were first seen by the service was similar for both younger and older men. The majority of men were seen within 48 h or at a time suitable to them soon after their referral, with only a small majority being seen later for other reasons (e.g., therapist availability, men’s work commitments). There were no significant differences on core measures related to the variation in the speed at which men were first seen at the service (*p* > 0.94). Most of the young men were white British (73%), single (63%), living with family (20%), and employed (34%). One third (34%) of the young men were seen within 48 h of their referral. Younger men were less exposed to suicidality within their lives compared to older men (30% vs. 39%). Both, younger and older men had similar histories of suicide attempts or self-harm (75% vs. 74%). Baseline characteristics are given in Table 1. In terms of ethnicity, relationship status, sexual orientation, employment status, and the CORE 34 clinical outcomes measure, no significant differences were noted for both groups. With regard to the types of services men were referred from prior to attending James’ Place, the majority of referrals came from secondary (37%) and primary care (23%). The proportion of men referred from each type of service does not differ by age (*p* > 0.46).

### 3.1. CORE-34 Clinical Outcome Data (CORE-OM)

For all subscales of the CORE 34, there was a statistically significant reduction in mean scores between assessment (*n* = 322) and discharge (*n* = 129) (F (1) = 571.75, *p* ≤ 0.0001, partial eta squared = 0.80), demonstrating a large effect size (Table 2). There was a clinically significant change for 39% of men using the service, with mean scores reducing by 10 or more, indicating a level of distress classed as healthy. Two percent of men demonstrated a reliable change with a reduction of five or more in the clinical distress scores following therapy, and 2% showed no clinical change. No significant differences were reported between younger and older men on distress scores (F = (2, 140) 1.55, *p* > 0.05), either at initial assessment (*p* > 0.05) or discharge (*p* > 0.05), but younger men showed lower levels of distress at initial assessment and lower levels of wellness than older men at discharge. However, it is worth noting that the mean score for all age groups fell within the severe distress category of the CORE-OM at assessment and mild or healthy levels at discharge. There was no discharge assessment score for 57% of the men who were engaged in the service due to some not attending their final sessions and others not completing the questionnaire following their final session. However, this did not indicate that men were not engaged in the therapy provided by the service.

### 3.2. Precipitating Factors for Men Help-Seeking in Suicidal Crisis

Table 3 shows the precipitating factors related to the current suicidal crisis for the men help-seeking that were recorded by the referrer or James’ Place for men who were self-referring. For young men, the most commonly reported factors were relationship breakdown (*n* = 43), family problems (*n* = 34), university (*n* = 24), work (*n* = 23), bereavement (*n* = 21), and debt (*n* = 18). Older men had similar or higher levels of precipitating factors than younger men for all except university stress (15% vs. 1%). There was no relationship between the precipitating factors and the levels of general distress found at the initial assessment (*p* > 0.05). There were no significant differences in general distress between those with and without each precipitating factor (*p* > 0.05) at the initial or discharge assessment and no significant relationship between any of the precipitating factors and distress scores (*p* > 0.05).

The psychological factors significantly affecting older men compared to younger men at initial assessment were entrapment (46% vs. 62%; *p* = 0.02), defeat (33% vs. 52%; *p =* 0.01), not engaging in new goals (38% vs. 47%; *p* = 0.02), and positive attitudes towards suicide (14% vs. 18%; *p* = 0.001). At discharge assessment, older men were significantly more likely to have an absence of positive future thinking (15% vs. 8%; *p* = 0.03), have less social support (45% vs. 33%; *p* = 0.02), and feelings of entrapment (17% vs. 14%; *p* = 0.02). Both younger and older men were commonly affected by rumination (77% vs. 78%; *p* = 0.82), past suicide attempts or self-harm (75% vs. 74%), thwarted belongingness (71% vs. 71%; *p* = 0.40), humiliation (51% vs. 67%; *p* = 0.64), and impulsivity (44% vs. 51%; *p* = 0.95) (see Appendix A: Table A1).

### 3.3. Engagement with Therapy

For both older and younger men, the mean number of sessions engaged with therapy was six, ranging between 1 and 18 sessions. Younger men completed the discharge assessment more compared to older men (64% vs. 59%). However, there were no significant differences (*p* > 0.05). Younger men were less likely to be referred onward to another service (7% vs. 15%). Both groups were most commonly referred to a psychological talking therapy for men, and older men were also referred to addiction and debt services.

## 4. Discussion

### 4.1. Main Findings

To our knowledge, this is the first study exploring help-seeking and engagement by young men in a suicidal crisis. Men who engaged with the innovative targeted community-based therapeutic service showed a significant reduction in general distress from assessment to discharge. There were no significant differences in the engagement or therapeutic effectiveness of the model between younger and older men. The findings relating to the psychological, motivational, and volitional factors offer further support for the utility of the IMV model [47,48], CAMS [46], and Joiner’s [45] model for understanding suicidal behaviour, as men reported difficult life circumstances prior to their suicidal crisis. Young men commonly reported many of the key factors in these models at the time of their suicidal crisis (e.g., feelings of defeat, entrapment, thwarted belongingness, hopelessness, humiliation, social isolation, and experiences of rumination); however, these factors were significantly reported for older men. With regard to precipitating factors of the suicidal crisis, our research supports that social aspects that increase suicide risk, particularly for young men, such as relationship breakdown and family problems [2,57,58,59,60], are the most common factors within our sample. Both groups of men engaged with therapy at similar levels and on average attended for six sessions (range 0–18). Younger men were less likely to be referred onward to another service than men over 30 years. At discharge, older men were more likely to have an absence of positive future thinking, have less social support, and have feelings of entrapment than younger men, suggesting that younger men may have benefited more from the therapy. Overall, the study has demonstrated the benefits of a rapid access tailored intervention, particularly for young men in suicidal crisis.

### 4.2. Strengths and Limitations

This research has a number of key strengths, with James’ Place being the first community based therapeutic suicide prevention centre in the UK. Previous studies [4,8,9,10,11,12,13] have typically been retrospective and included information from third parties, such as bereaved family members; this quantitative prospective study accessed information about young men at the time of their suicidal crisis. Its novel and timely findings can inform future service implementation to reach a male population group that is at high risk of suicide [41] and less likely to seek help [15], thus filling an important gap in service provision that traditional care pathways are not always able to reach.

A further strength of this study is the light it sheds on the specific precipitating factors leading young men without pre-existing mental illness into a suicidal crisis. The present findings point to the importance of informing/educating wider stakeholders, such as the general public, workplaces, military services, schools/universities, and GPs, about community-based services that can help to reduce suicidal thoughts and behaviour in men. Similar to previous studies [4,7,14,23], the findings emphasise that help is required that goes beyond the medical model, as many of the reported factors that led to the suicidal crisis in this group of young men were relationship or family breakdowns, university stress, and debt. One out of five of the presenting young men at the service were students from local universities, thus highlighting the risk of this vulnerable at-risk population and the need for tailored interventions within higher education institutions [4].

However, these findings should be interpreted in the context of some methodological limitations. The first issue is that of missing data. Whilst this is to be expected due to attrition and establishing processes in the first few years of running a new service, it has been a valuable learning point for improving the service going forward. Some data were reliable on men completing and returning questionnaires, others on referrer documentation, and, finally, others on therapists recording psychological, motivational, and volitional factors during therapy sessions. Inconsistency in how these data were collected has been highlighted. Having monitoring and evaluation built into the service from the start has enabled timely evidence and data to be fed back, but this aspect still requires improvement. This had led to the implementation of clinical data systems, thus providing evidence for the need of funding a costly resource to improve data collection.

With regard to sampling, it is important to note that only records for men who were seen by the service were sampled; therefore, the results may not reflect the information for men who did not have contact with the service who may also have been in suicidal crisis. Thus, it is difficult to draw firm aetiological conclusions from these data. It is not uncommon for men not to engage with therapy [21,22,23], but for this study, it does possibly limit the findings to those who engaged with therapy and those who were satisfied with the intervention. For future research, it would be good to know more about the men who did not engage with the intervention fully. This, however, was a deliberate decision in the design phase of this study, as one of the main aims was to examine the pilot stage of feasibility of the service for younger and older men. This was to ensure that the relevant population of men were being reached and referred to the service, and that the service being provided was efficient and safe for men in helping to reduce their suicidal distress. Due to the significant reduction in clinical risk for most of the men of all ages who used the service, we think these findings are even more striking.

Another limitation is the absence of a control group. Comparative data would highlight how the outcomes of these men compare to men receiving other or no treatment (or those that drop out of treatment). However, due to ethical issues around the safety of people placed in a control group, comparator studies are more difficult to conduct within suicide research [61].

This study was conducted in a service in North West England. Therefore, care must be taken when attempting to generalise these findings to other geographical regions. This region is reported to have the highest rate of suicides in the UK [3], which may have influenced the study findings when comparing to regions where the suicide rate is much lower. The higher rates of suicide may be reflective of the health inequalities reported by the Public Health England [PHE] report [62]. The life expectancy across this region is lower compared to that of most of England, thus increasing the importance of such interventions. Previous research has demonstrated that the provision of community-based services for those in suicidal distress is lacking [26,36,37,38]. The findings of the current study support that this type of service provision within a community setting can play a significant role in reducing suicidality for men.

## 5. Conclusions

Our results support the use of the James’ Place model for men in suicidal distress to aid in potentially preventing suicides in this high-risk group of the population and highlight the heightened distress among university students. A move away from the traditional medical model and the implementation of community-based tailored crisis services for men should be an essential part of any suicide prevention strategy. Future research needs to assess the long-term effects of the model for young men in order to understand whether the effects of the therapy are sustainable over a period of time following discharge from the service.

## Figures and Tables

**Figure 1 ijerph-18-07319-f001:**
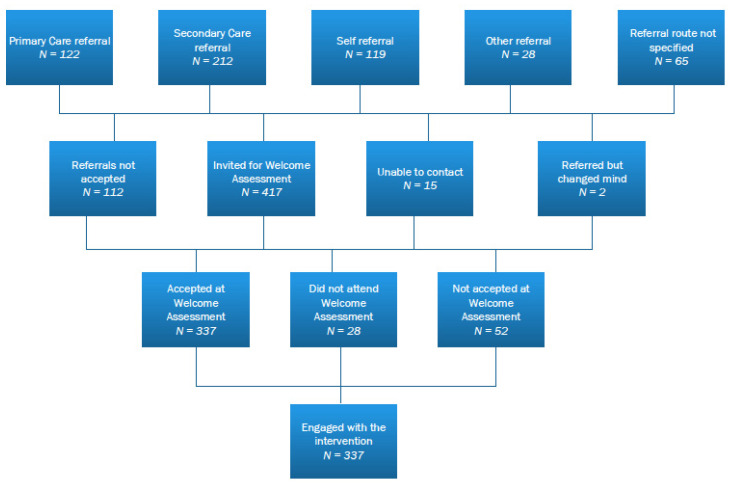
Flow diagram of the referral for men using James’ Place in years 1 and 2.

**Table 1 ijerph-18-07319-t001:** Demographic characteristics of the men help-seeking at James’ Place.

Demographic	18–30 Years	31+ Years	Significance between Groups against CORE-OM
	*N* = 161 (%)	*N* = 176 (%)	
Ethnicity			*p* = 0.80
White British	116 (72)	140 (79)	
Other	26 (16)	10 (6)	
Missing	19 (11)	26 (14)	
Relationship Status			*p* = 0.84
Single	101 (63)	66 (38)	
Married	0 (0)	36 (20)	
In a relationship	10 (6)	10 (6)	
Divorced	0 (0)	6 (3)	
Separated	1 (1)	12 (7)	
Widowed	0 (0)	1 (1)	
Missing	49 (30)	45 (25)	
Sexual Orientation			*p* = 0.32
Heterosexual	35 (22)	35 (81)	
Homosexual	5 (3)	7 (4)	
Bisexual	2 (1)	1 (1)	
Missing	119 (74)	133 (76)	
Employment Status			*p* = 0.94
Employed	54 (34)	73 (42)	
Unemployed	41 (26)	50 (28)	
Students	33 (21)	14 (8)	
Missing	33 (20)	39 (22)	
Referrer			*p* = 0.46
Secondary Care	57 (35)	66 (38)	
Primary Care	42 (26)	35 (20)	
Self-Referral	28 (17)	45 (26)	
Other	7 (4)	12 (7)	
Not specified	27 (17)	18 (10)	

**Table 2 ijerph-18-07319-t002:** Descriptive data for Core OM measures by age group.

CORE 34 Measure	18–30 Mean	18–30 SD	31+ Mean	31+ SD	Significance between Groups against CORE-OM
Initial Distress (*N* = 322)	85.30	17.17	87.47	18.34	*p* = 0.47
Discharge Distress (*N* = 129)	37.61	22.09	32.21	23.33	*p* = 0.16

**Table 3 ijerph-18-07319-t003:** Precipitating factors for men help-seeking in suicidal crisis.

Precipitating Factor	18–30	31+	Significance between Groups against CORE-OM
	(*N* = 161)	(*N* = 176)	
Relationship breakdown	43	40	0.13
Debt and Financial issues	18	38	0.40
Family problems	34	45	0.16
University stress	24	2	0.65
Work stress	23	32	0.36
Bereavement	21	34	0.07
Mental health	11	11	0.41
Drug Misuse	10	9	0.45
Alcohol misuse	10	12	0.81
Victim of past abuse/trauma	9	27	0.33
Legal problems	6	9	0.20
Perpetrator of a crime	5	3	0.22
Gambling	3	5	0.91
Housing issues	5	7	0.18
Physical health	5	14	0.48
Victim of bullying	4	4	0.19
Sexuality	5	3	0.12
Victim of crime	2	5	0.83
Bereavement by suicide	3	7	0.99
Relationship problems	4	9	0.78
Concerns about others health	2	0	0.58
Related to COVID-19/lockdown	2	6	0.46
Caring responsibilities	0	3	0.70
Other	0	2	

## Data Availability

Restrictions apply to the availability of these data. Data was obtained from James Place Liverpool and anonymised datasets are available from the authors with the permission of James Place Liverpool.

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
