# Peer review of "A Case Series Study of Help-Seeking among Younger and Older Men in Suicidal Crisis"

_ijerph, 2021, doi:10.3390/ijerph18147319_

Round 1

Reviewer 1 Report

Please see attached

Author Response

We thank you for your comments and have attached our response in the attached document.

best wishes

Pooja

Reviewer 2 Report

Thank you for an opportunity to review a mss reporting on a case series study of help-seeking among younger and older men in suicidal crisis attending James’ Place in the UK, a community-based service delivering a clinical intervention for men. The paper is well-written and reports on psychosocial interventions for men in a real-life setting. The authors have clearly and correctly identified strengths and limitations of their work.

A few questions and suggestions for the authors:

  1. Can the authors clarify the aims of the study? Line 168-169 stipulates that the study aims to "compare the differences between help-seeking, engagement and outcomes for younger and older men", while the review of the literature focuses on younger men and the section on main findings starts with a statement regarding young men (line 343).
  2. Participants: Can the authors specify eligibility criteria for inclusion in the reported study and analysis?
  3. Clinical records: Can the authors provide more information on data included in clinical records?
  4. Line 209: Can the authors provide more information on "assessment by the therapist during each session"? Lines 215-223 are not clear and mention "secondary outcomes". I am not sure what this point refers to.
  5. Please, correct typos in the mss, eg "men’s work commeitments" (line 269), "did not indictae that" (line 301), "It’s novel" (line 371).

Author Response

We thank you for your comments and have attached our response in the attached document.

best wishes

Pooja

This manuscript is a resubmission of an earlier submission. The following is a list of the peer review reports and author responses from that submission.

Round 1

Reviewer 1 Report

This is a valuable study looking at a resource for men in crisis. The model for the support service is simple and the results indicate an effective service for men. This article will be important evidence in supporting further funding for similar projects around the world.

The results add to the literature around suicide by men and the social factors that impact young men specifically in comparison to older men in crisis. The methods and analysis section are clear. The study highlights the effectiveness of this resource well and the results are clearly presented. I feel the study highlights the effectiveness of this resource well and the results are clear. Overall a valuable contribution to suicide prevention for young men.

Minor points:

The introduction is slightly formal, a more sympathetic tone to the impact of suicide may be more appropriate.

Line 69 – would avoid using “completed suicide”.

The introduction has a lot of references to studies describing cohorts of men and their experiences, many of these studies were conducted several years ago. These references may still be relevant and important to men - but it is not clear, at times, where you are describing young men of today. The generational change has been noted but could be a clearer message for the reader.

The patient and public involvement section feels slightly too formally written and repetitive. This is such an important section demonstrating how experiences and expertise worked together to develop the service but would benefit from simplification for the reader.

Reviewer 2 Report

  1. The title and the purpose of the study and the content of analysis are inconsistent.
  2. Among the many characteristics of the participants, why did you include only one factor, age, as independent variable.
  3. I don't think you properly compared the differences between help-seeking, engagement and outcomes for younger and older men in this study. Table 3 compared the main outcomes of the two groups, but the analysis was crude. Table 4 cannot be seen as comparing the help-seeking and engagement of the two groups.
  4. The tables are very fragmented and do not contain all of the necessary information. (For example, there is no p value in Table 3)
  5. I think the analysis is too crude and, on the other hand, the description of the introduction and discussion overstated the meaning of the analyzed content.

Reviewer 3 Report

Thank you for the opportunity to review the paper ‘Help seeking and engagement for young men aged 18 to 30 years in suicidal crisis: a prospective cohort study’. The study presents an interesting examination of the effectiveness of a suicide prevention intervention for men. This is important work, and addresses a gap in the knowledge about gender sensitive suicide prevention interventions. However, I do have some suggestions for revision mostly centred on the need for more detail information in the methods section about data collection methods, improved clarity of the results in relation to the data collection methods, and a linking of the introduction and discussion to other research regarding factors precipitating men’s suicidal crises.

INTRODUCTION

In first paragraph please qualify – ‘the rate of suicide among young men has….’ with the location of the statistics. I don’t believe this is a global figure as is implied.

On page 2 in the third paragraph the sentence ‘gender role socialisation typically imposes higher thresholds for males than females 76 in terms of emotional expression of sadness, insecurity and vulnerability’ is unclear in its meaning.

The aims at the end of the introduction are unclear. Effectiveness for what? To reduce suicidal risk?

MATERIALS AND METHODS

The James’ Place Model

This section does not describe materials or methods of the research, but the method of the intervention, and would be better situated in the introduction. In this section it would also be good to know how the intervention was tailored to men (i.e. gender sensitive – if it is) to overcome some of the issues raised in the introduction.

CORE-34 Clinical Outcome Measure (CORE-OM)

More information about how, when, and by whom the CORE-OM administered to participants.            

Assessment of psychological, motivational and volitional factors

It is unclear how ‘a range of psychological, motivational and volitional factors’ were assessed. Was this by the anonymised questionnaire? What questions were asked? When and how was this questionnaire administered? It would also help the reader to have these terms explained.

How do you know that ‘clients felt comfortable and at ease’ – is this the judgement of the researcher?

Did therapists collect data? If so, I imagine that there is some bias there. How many therapists collected the data? How was consistency in data collection obtained?

Data analysis

Here the clinical record is mentioned, but this is not mentioned previously. This information should be provided in the materials section (for which there should be subheading). The data extracted from the clinical record, and the method for doing so, should be described.

Patient and public involvement

Some of this description of the Centre would be better placed in the ‘the James’ Place Model’ in the introduction.

RESULTS

I am not sure that the speed with which men were first seen contributes to addressing the paper’s aim of exploring effectiveness. This could be omitted.

It is unclear which results pertain to which data source. Results are not clearly presented regarding psychological, motivational and volitional factors.

DISCUSSION

I think it is a bit of stretch to say that attendance showed a significant reduction in general distress given that there was not a discharge score for 57% of men – keeping in mind also that in your introduction you point out that men often disengage from services due to dissatisfaction with the service. Your logic would suggest that those that didn’t complete the discharge score because they dropped out, may not have had a great experience.

It is not clear how the findings relate to the psychological, motivational and volitional factors. This link should be made more explicit to the reader and map more closely to the method section.

There have been other international studies that have looked at factors precipitating suicidal crisis in men – these should be mentioned in the paper – in either or both the introduction and discussion.

Limitations

The study sheds light on the precipitating factors leading young men without pre-existing mental illness into a suicidal crisis 

Another limitation is the high number of men who did not complete a discharge assessment and did not complete the intervention – this is not uncommon but does limit the findings possibly to those who were satisfied with the intervention. It would be good to know more about the men who did not engage with the intervention fully.

Another limitation is the absence of a control group – so it is not known how these men’s outcomes compare to men receiving other, or no, treatment (or those that drop out of treatment).